# Effects of an automated digital brief prevention intervention targeting adolescents and young adults with risky alcohol and other substance use: study protocol for a randomised controlled trial

Pia Kvillemo [1], Anna K Strandberg,[1] Johanna Gripenberg,[1] Anne H Berman,[2,3] Charlotte Skoglund,[1,4] Tobias H Elgán[1]

For numbered affiliations see end of article.

**Correspondence to**
Dr Pia Kvillemo;
pia.kvillemo@ki.se

## ABSTRACT

**Introduction** Adolescence and young adulthood is a period in life when individuals may be especially vulnerable to harmful substance use. Several critical developmental processes are occurring in the brain, and substance use poses both short-term and long-term risks with regard to mental health and social development. From a public health perspective, it is important to prevent or delay substance use to reduce individual risk and societal costs. Given the scarcity of effective interventions targeting substance use among adolescents and young adults, cost-effective and easily disseminated interventions are warranted. The current study will test the effectiveness of a fully automated digital brief intervention aimed at reducing alcohol and other substance use in adolescents and young adults aged 15 to 25 years.

**Methods and analysis** A two-arm, double-blind, randomised controlled trial design is applied to assess the effectiveness of the intervention. Baseline assessment, as well as 3-month and 6-month follow-up, will be carried out. The aim is to include 800 participants with risky substance use based on the screening tool CRAFFT (Car,Relax, Alone, Forget, Friends, Trouble). Recruitment, informed consent, randomisation, intervention and follow-up will be implemented online. The primary outcome is reduction in alcohol use, measured by Alcohol Use Disorders Identification Test total score. Secondary outcomes concern binge drinking, frequency of alcohol consumption, amount of alcohol consumed a typical day when alcohol is consumed, average daily drinks per typical week, other substance use, mental health, sexual risk behaviours and perceived peer pressure. Moreover, the study involves analyses of potential moderators including perfectionism, openness to parents, help-seeking and background variables.

**Ethics and dissemination** The study was approved by the Swedish Ethical Review Authority (no. 2019–03249). The trial is expected to expand the knowledge on digital preventive interventions for substance using adolescents and young adults. Results will be disseminated in research journals, at conferences and via the media.

### Strengths and limitations of this study

► A double-blind, randomised controlled trial is considered to be the most robust experimental design, enabling causal inferences, controlling for selection bias and participant allocation bias.
► An active control condition, blinded to the study condition, controls for expectation, detection and performance bias.
► Analysis of several potential moderators, that is, perfectionism, openness to parents, help-seeking and background variables will contribute to the understanding of possible effects.
► A convenience sample may reduce generalisability due to possible selection bias at recruitment.

**Trial registration number** 24 September 2019, ISRCTN91048246; Pre-results.

## INTRODUCTION

Substance use is a major public health concern causing individual suffering as well as societal costs.[1–3] Substance use in childhood and younger ages is particularly harmful since the brain is undergoing critical development during this time in life, which makes it more vulnerable to addictive substances.[4 5] In adolescence, some individuals are especially prone to various risk behaviours, increasing the risk of, for example, substance use.[6] Additionally, having substance using peers is a prominent risk factor for one's own substance use at this age, since peers are increasingly important in teenagers' social life.[7] Thus, the initiation of substance use often occurs during this period.[6] After initiation, the use of substances frequently increases during adolescence and young adulthood, posing

BMJ

a number of risks for the individual.[2 8 9] Early onset of substance use implies a risk of severe adverse effects on psychological and physiological development.[2 8 9] Hazardous use of alcohol, cannabis and other substances can lead to chronic problematic consumption patterns and addiction that can significantly influence the developmental trajectory during the transition from childhood to adulthood.[10] More acute consequences include, for example, mental or psychiatric problems, problems of academic adjustment,[11] accidents[12] and problems with the police or legal authorities.[13 14] Risk behaviours, such as substance use, binge drinking, intoxication and sexual risk behaviour often co-occur, and the use of illicit substances, is for example, often accompanied by alcohol consumption.[1 2 11 15–19] Polysubstance use, that is, simultaneous use of different substances, may imply an increased risk of various negative outcomes, for example, mental health problems, shown to increase in magnitude and over time, with the number of substances used.[20] The well-established association between substance use and mental health problems is bidirectional, sometimes manifested in self-medication for mental health problems, and sometimes as an increase in such problems due to substance use.[21] While most of prior research shows that low socioeconomic standard is a risk factor for substance use and related problems,[22 23] the current study is informed by recent research on mechanisms involved in adolescence alcohol and other substance use among upper secondary school students from affluent areas, showing associations between internalising symptoms and substance use,[24 25] as well as associations between achievement pressures (particularly excessive perfectionistic strivings) and isolation from parents (particularly low perceived closeness to mothers).[24 26] The latter finding is in line with previous research showing that a positive parent-child relationship is a protective factor related to lower likelihood of adolescence substance use.[27 28]

Although alcohol use has declined among adolescents during recent years, some individuals drink more and binge drinking is still highly prevalent among young adults, where the largest proportion of alcohol risk consumers is still found.[29 30] Moreover, a high prevalence of substance use and polysubstance use among young people in Europe was recognised in the European Union Drug Strategy 2005 to 2012, which calls for action.[31] Also, established cannabis users among Swedish upper secondary school students tend to use cannabis more often than before, according to recent survey data.[32] In light of these circumstances, it is of utmost importance to target and tailor attractive preventive interventions for adolescents and young adults. One way to reach large groups of young people is to disseminate digital interventions online (ie, via the Internet). In Sweden, 98 per cent of the population has access to the Internet at home and virtually 100 per cent of 16 to 25 year olds use the Internet in mobile phones, tablets or computers.[33] Digital interventions have several advantages over traditional ones. They may reduce the stigma around risk behaviours, they are accessible at any time and place and can be cost-effective because minimal staff resources are used for implementation, especially if they are fully automated.[34 35] Digital interventions may thus be particularly suitable for adolescents and young adults.[36] To date, digital prevention programmes targeting alcohol and other substance use among adolescents and young adults that meet scientific evaluation criteria are scarce. An early example is eScreen.se, offering screening for alcohol and drug use with personalised feedback.[37] A more recent example is a fully automated brief motivational intervention for substance-using 16 to 18 year olds, tested in an randomised controlled trial (RCT) in four European countries,[38–40] where our research group was one of the partners. The intervention, named WISEteens, relied on Motivational Interviewing (MI)[41] and models of Social Influence.[42 43] The goal of the MI approach was to enhance motivation to change by exploring and resolving ambivalence about substance-related behaviours. An important element in the intervention was personalised feedback about one's substance use behaviours in relation to normative comparisons, as personalised feedback is perceived as more relevant for changing behaviour than more general information.[44] The normative feedback included information about how a specific reference group actually consumed substances, in order to correct participants' 'inflated perception',[45] which has proven effective for reduction of alcohol consumption in young adults in previous meta-analyses.[46 47] For adolescents and young adults, a normative feedback approach may be particularly appealing, assuming they are curious about how their substance consumption compares to their peers.[48] Likewise, an important element of the intervention was a focus on substance-related social norms and training on how to avoid social high-risk situations and how to resist peer pressure; that is, raising refusal self-efficacy.[49] In line with the social influence hypothesis stating that a prominent risk factor for substance use in adolescence is the influence of substance using peers,[7] targeting peer pressure in brief digital interventions has been an important element for reducing alcohol use among adolescents.[50] WISEteens demonstrated significant between-group effects for alcohol use, indicating that a targeted brief motivational intervention in a fully automated digital format can be effective to reduce drinking and lower barriers for accessing substance use service in hazardous drinking adolescents.[38]

Digital interventions, often delivered online, generally offer adaptations of evidence based face-to-face interventions, for example, MI. The effects of MI on several health-related behaviours have been evaluated in a number of other studies, especially among adults.[51] Among adolescents, systematic reviews and meta-analyses on MI for various health-related behaviours show mixed results. A review of six studies of MI for reducing alcohol consumption in the emergency room setting suggested that MI was at least as effective as other brief interventions in the same setting.[52] Another review of 24 studies of different

brief interventions for reducing alcohol consumption and alcohol-related problems among adolescents showed a significant effect that persisted up to 1 year compared with control interventions, with greater effects for MI. Effects were consistent over diverse settings and particularly effective components included decisional balance (juxtaposition of pros and cons of change)[53] and goal-setting.[54] Regarding MI for illicit drug use among adolescents, a review of 10 studies showed no effects on drug use behaviours; however, changes in attitudes towards drug use were found, which could be translated into intentions to change behaviours.[53] A scoping review concerning MI for reducing sexual risk behaviours among adolescents identified 29 unique studies with varying designs and conceptualisations of MI and specific risk behaviours, making it difficult to generalise regarding outcomes but indicating the need of more research.[55] MI has most commonly been delivered as an individual face-to-face intervention,[51] but has also been provided in other forms, for example, via telephone or digitally, with various results on, for example, substance use behaviours.[56 57] Digital programmes based on MI have demonstrated effect in the form of reduced alcohol consumption among young adults.[58 59] Additionally, a combination of screening with a short intervention has shown similar effects on alcohol consumption among adolescents.[60] Brief motivational interventions are empirically supported individual level interventions for reducing alcohol consumption.[45 61] Such interventions have in the last decades also been digitally provided.[62 63] Digital interventions can be provided with or without human guidance, that is, more or less automated.[64] Previous research has demonstrated that even fully automated interventions can reduce alcohol-related problems for young people with risky alcohol consumption up to 12 months after implementation,[59 65] and indicated potential effect on cannabis use in certain groups.[66 67] Although various digital substance use prevention interventions have been developed and tested with promising results,[38 68] studies on the effects of fully automated digitally delivered and MI-based substance use prevention programmes for adolescents and young adults are still few[68] and more well-designed studies are warranted in order to obtain evidence for potential effects.

## Aim

The study described in this protocol builds on results previously obtained in a European study of a digital brief MI-based substance use prevention intervention (WISE-teens), delivered online to adolescents in a fully automated form.[38] In this study, the effectiveness of a modified version of this intervention, with new graphics and population based feedback-generating data on alcohol consumption covering all included age groups, will be tested across a range of outcome measures among 15 to 25 year olds with hazardous alcohol or other substance

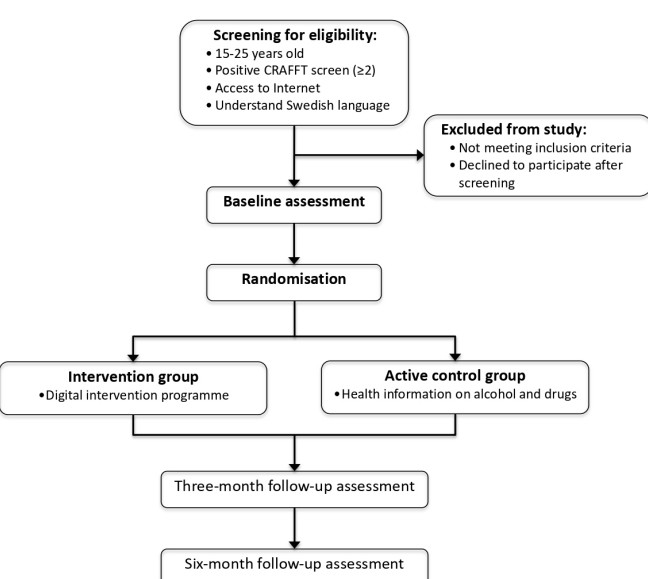

**Figure 1** Study design. The CRAFFT is a screening instrument for use of alcohol and other substances and stands for the keywords Car, Relax, Alone, Forget, Friends, and Trouble.

use. The primary aim is reduction in frequency and quantity of substance use.

## Rationale and hypothesis

The current study will add to existing evidence regarding the effectiveness of digital interventions aimed at reducing alcohol and other substance use among young people. We hypothesise that participants in the intervention group will report a significant reduction in substance use (primarily alcohol use), compared with an active control group receiving health information about various substances. A number of potential moderators will be assessed to confirm or contradict previous research on factors influencing the development of substance use patterns over time, and effects of prevention interventions in different target groups.

## METHODS AND ANALYSIS

In order to evaluate the effectiveness of the digital screening and brief motivational intervention, a double-blind, two-arm, RCT study design is planned with baseline assessment at study entry and follow-up assessment at 3 and 6 months across a number of outcome variables. Figure 1 displays the trial design. We will also explore and test for moderator effects. To outline and report the current study we used Standard Protocol Items: Recommendations for Interventional Trials reporting guidelines.[69]

## Recruitment

The target group for the current study is adolescents and young adults aged 15 to 25 years with self-reported substance use, a capacity to understand the Swedish

language and access to the Internet. Based on a power calculation to detect a small effect size (Cohen's $d$=0.2),[70] the aim is to include 800 participants.[71] The participants will be recruited during January 2020 to October 2020 by offline marketing at youth health clinics, other health-care units for youth and at upper secondary schools in Stockholm County, as well as by social networks, using online banner advertisements. This will result in a convenience sample from the general Swedish population.[44] Results from follow-up measures are to be completely received in 2021. If sufficient number of participants have not been recruited according to plan, we will extend the recruitment period and intensify the advertising by increasing visibility on social platforms. To enhance participation and follow-up rates, an incentive in the form of a movie ticket, with a value up to 15 Euros, for each completed follow-up assessment will be provided, that is, up to three tickets. Up to three reminders will be sent out if participants do not respond to follow-up assessments.

### Screening and informed consent

Potential participants will be guided to an online landing page with screening by an adapted version of the screening tool for use of alcohol and other substances, CRAFFT (Car, Relax, Alone, Forget, Friends, Trouble).[72] This six-item screening tool has demonstrated criterion validity and appropriateness for identifying substance-related problems among adolescents.[73–75] The primary eligibility criterion for participating in the study will be a score of 2 or more on the CRAFFT, since this value has shown satisfactory sensitivity for identifying substance use problems.[72] Those who score 2 or higher will be offered participation in the study and given information on confidentiality, voluntariness of participation and human subject protections.[44] They will also be provided with contact information on suitable counselling service providers. Those who agree to participate will be asked to give digital informed consent.

### Measures

All assessments will occur online. The selection of measures and associated instruments is based on the aim of the study and the theoretical base concerning factors influencing substance use. Based on results from the previous European WISEteens study,[38] showing significant effects on alcohol consumption but not on other substance use, the primary outcome is alcohol use, measured at 3-month and 6-month follow-ups. Secondary outcomes include binge drinking, frequency of alcohol consumption, amount of alcohol consumed a typical day when alcohol is consumed, average daily drinks per typical week, at 3-month and 6-month follow-ups, other substance use, mental health, sexual risk behaviours and perceived peer pressure at 3-month and 6-month follow-ups. The moderating variables include perfectionism, openness to parents, help-seeking due to mental health problems

and socio-demographic and personal characteristics, (ie, sex, age, residence, occupation, school performance (for students) and parents' education).

Alcohol use, including the primary outcome, will be measured using two instruments. One of them is the short version of the Alcohol Use Disorders Identification Test (AUDIT-C),[76] measuring frequency and amount of consumed alcohol and frequency of binge drinking, providing a widely used and valid index sum score for problematic alcohol use among adolescents.[75 77] The other is the Daily Drinking Questionnaire (DDQ),[78] measuring a variety of parameters of alcohol use in a typical week. Previous research has demonstrated that the DDQ is highly correlated with other measures of self-reported alcohol consumption.[79] The primary outcome will be measured by AUDIT-C total score. The two measures of alcohol use complement one another in that the AUDIT-C primarily offers an indication of hazardous or harmful use,[80] whereas the DDQ quantity measure (average daily drinks per typical week) will indicate the level of alcohol consumption in grams per week. Other substance use will be measured by a short version of the Drug Use Disorder Identification Test (DUDIT),[81] including the first four items (DUDIT-C) which assess frequency of consumption of drugs other than alcohol, frequency of different types of drugs other than alcohol used at the same occasion (1 = never to 5 = *four or more times a week*) and number of occasions when drugs other than alcohol are consumed on a typical day of drug use (1 = zero to 5 = *seven or more*). The DUDIT has been found to be effective in screening for drug-related problems in clinically selected groups,[82] and both the DUDIT and the DUDIT-C have proven useful in the context of public health surveys in Sweden.[81 83] Mental health will be measured by The WHO-5 Well-Being Index, which has demonstrated validity both as a screening tool for depression and as an outcome measure in clinical trials.[84] This scale has been successfully applied across a wide range of study fields, translated into more than 30 languages, including Swedish,[85] and found psychometrically sound. Changes in sexual risk behaviours will be measured by multiple choice questions on sexual behaviour under the influence of alcohol and/or other substances and unprotected sex, previously used in a survey among Swedish visitors at youth health clinics in Stockholm County.[86] Changes in perceived peer pressure will be measured by two items retrieved and adapted from the peer pressure inventory.[87] Perfectionism will be measured by two subscales of the Frost Multidimensional Perfectionism Scale.[88 89] Openness to parents will be measured among participants who are up to 20 years old with questions about disclosure from Stattin and Kerr,[90–92] slightly modified into statements to be compatible with the digital online format. Help-seeking due to mental health problems will be measured by dichotomous questions regarding if, and to what healthcare provider, the participant has turned for help. Additionally, background data will be asked for using multiple choice questions: sex (man, women, other), age (15, 16…25), residence (name

of municipality), occupation (secondary school, upper secondary school, university, not student, working, practicing, unemployed, other), school performance (merit value for students) and parents' education (secondary school, upper secondary school, university, don't know).

### Randomisation

The study is double-blind, thus neither the participant nor the researchers will know which participant is allocated to the intervention or to the active control condition. After baseline assessment, the participants will be automatically randomised to one of the two study groups by a computer programme using an unrestricted randomisation protocol. Participants will then be informed about the name of their programme and given access immediately.

### The intervention

Participants allocated to the intervention group will initially be asked to give additional information on alcohol and other substance use and to state their body weight, in order to generate personalised feedback on substance use behaviour and to estimate alcohol blood concentration on a typical drinking occasion.

#### Description of the intervention content

The intervention is interactive, digitally delivered online in a fully automated form and requires approximately 20 min to complete. Personalised feedback is given to the participants based on their responses to previous assessment and suggestions on how to respond to this feedback are provided. This interactivity imitates a face-to face 'dialogue' with techniques from MI such as an empathic approach, rolling with resistance, aiming at creating a dissonance between actual and desired behaviour, raising self-efficacy and at the same time avoiding argumentation.[57] The intervention consists of three main components outlined below, and additional health-related information.

#### Personalised feedback

The personalised feedback includes an estimation of the participants' blood alcohol concentration and information on the associated risks concerning the participants' heaviest drinking episode during the past 30 days. The value will be based on a measure of Peak Drinking Quantity[93] and estimated using the Widmark formula, which takes into account weight and sex.[94] The participants will receive graphed feedback regarding number of standard alcohol units per week that they think their peers consume (descriptive norms), as well as the participant's individual levels of consumption in relation to comparative data (actual drinking levels) from a reference group. Comparative data (AUDIT-C scores) will be taken from alcohol prevalence estimates found in a nationally representative sample of 16 to 25 year olds undertaken by the Public Health Agency of Sweden. Comparative feedback will be available for drinking but not for substances other than alcohol.

#### Interactive MI-based exercises

The exercises provided in the current intervention build on the assumption that participants may hold certain levels of ambivalence about their current substance use, and that if they are willing to make a change they may not know how to, or may not be confident that they are able to.[41] Therefore, the intervention uses importance and confidence rulers with a short summary and feedback to encourage participants to reflect on personal reasons for change and explore personal strengths and ability to change. Furthermore, the programme provides a decisional balance to pick up and graphically illustrate potential levels of ambivalence by offering the participants a list of possible pros and cons regarding the decision to change their current substance use.[57 95] Participants are instructed to choose statements that apply to them and are presented with the resulting balance sheet of their personal comparative potential gains and losses.

#### Practical advice

The intervention focusses on raising self-efficacy for being able to avoid drinking in social situations, if desirable.[50] The participant will be asked to select 3 among 12 provided drinking situations that they consider most tempting and rank them. The situations are adapted from the adolescent version of the Drinking Refusal Self-Efficacy Questionnaire (DRSEQ-RA).[96] According to the selection, a number of strategies will be offered for each of the selected drinking situations to provide participants with a tool kit necessary for engaging in and maintaining their behavioural goal.

#### Health-related information connected to substance use

Finally, the intervention programme includes health-related information associated with substance use. The information is provided optionally throughout the programme behind 'read more' buttons and also at the end of the programme. The information contains statements regarding risks connected to substance use that the participant can reflect on, as well as optional links for more information. There are also two frequently asked question sections about alcohol and cannabis, respectively, as well as information about how different substances may affect the individual, both physically and mentally. Finally, the information also contains cases, were the participant can read about some typical young persons who have used substances, why they used and what negative effects they have noted and why they chose to stop using.

### The control condition

The control group will receive the same general health-related information as the intervention group, that is, the additional information which is connected to the intervention programme.

## Statistical analysis

Randomisation checks of baseline variables regarding alcohol and other substances use and psychological state will be conducted using multivariate analyses of variance. To test the effectiveness of the intervention, we will assess whether participants in the intervention group to a greater extent report decreased substance use, sexual risk behaviours, perceived peer-pressure and improved mental health after 3 and 6 months, respectively, compared with participants in the active control group. Data analyses will consist of comparing outcome measurements with regards to within-group and between-group differences according to the intention-to-treat (ITT) principle in the primary analysis, accounting for all included participants regardless of whether or not they completed follow-up assessments. We will also perform per protocol analyses. The reason for choosing ITT in the primary analysis is the ambition to maintain the power of the study and also to avoid biassed results due to selection of those completing all follow-ups, as they may have special characteristics not representative for the whole study group, which might influence study results. The main analysis of effectiveness will use mixed effects regression models, which can be applied to both continuous and categorical outcomes and also non-normally distributed outcomes. Also, mixed-effects regression models are robust to missing data in longitudinal studies.[97] In the current study, we assume a quite large dropout rate, based on previous research. For example, a meta-analysis comprising 17 studies of unsupported (fully automated) interventions for depression showed 74% dropout rate at post-treatment.[98] Moreover, we expect data missing at random, which can be handled using mixed effects regression models. Separate models will be run to test each outcome, that is, alcohol consumption, binge drinking, frequency of alcohol consumption, amount of alcohol consumed a typical day when alcohol is consumed, average daily drinks per typical week, other substance use, mental health, sexual risk behaviours and perceived peer pressure at 3-month and 6-month follow-ups. Effect size will be calculated for the primary outcome variable (AUDIT-C total score). Potential moderators, including perfectionism, openness to parents, help-seeking due to mental health problems and socio-demographic and personal characteristics, (ie, sex, age, residence, occupation, school performance (for students) and parents' education) will serve as covariates in analyses of intervention effects. If moderator effects are found, we will carry out post hoc analyses stratified by the detected variable.

### Patient and public involvement statement

Patients and the public were not involved in the design and planning of the study, except for persons at the age of the target group reading manuscripts for the revised version of WISEteens and also participating in a pilot test.

## ETHICS AND DISSEMINATION

The current study was approved by the Swedish Ethical Review Authority (no. 2019–03249) and registered, pre-results, on IRCTN. Any important protocol modifications will be reported to IRCTN. For inclusion, all participants must give informed consent online prior to participation in the study. In order to participate, the participants will need to state a username (which may be fictitious or a pseudonym) and an email address. All data collection will be done without collecting personal identification information, only personal email addresses. The email address will be used to connect data from the baseline measurement to the follow-up measurements. At a later stage, the raw data file will be anonymised and each person assigned a number instead of the email address. The data will be stored in line with routines for handling and storing research data at Karolinska Institutet. All data is handled confidentially and will not be forwarded to third parties. Participating in the present study means that the participants need to reflect on their alcohol consumption and other substance use. In addition, questions will be asked concerning personal circumstances, including the participants' mental health and family relationships. These issues may be perceived as somewhat unpleasant. However, in the information that potential participants receive, they are informed that participation is voluntary and that they at any time can end their participation without explaining why. In addition, there will be reference to other types of official support (web pages and telephone numbers), if the advertisement or participation leads to concern about own substance use or related problems. Any issues brought up by the participants during the study will be documented and handled properly. Moreover, the research team includes professionals, such as a psychiatrist and a nurse, with possibility to refer participants to healthcare clinics if needed. The project's basic hypothesis is that the intervention will have positive effects, with regard to alcohol and other substance use among young people with risk use, and potential benefit may include decreased or ceased risk behaviours. Overall, the benefit for the research persons is considered to exceed any risk of discomfort. Results will be disseminated in scientific peer-reviewed journals, at conferences and via the media.

## DISCUSSION OF STRENGTHS AND LIMITATIONS

The current study has a number of strengths. The protocol describes a two-arm, double-blind, RCT, considered to be the most robust experimental design, controlling for selection bias and participant allocation bias and with the possibility to make causal inferences. Moreover, an active control condition, blinded to the study condition, controls for expectation, detection and performance bias. The collection and analysis of information on potential moderators, allowing for control of these factors, facilitates the understanding of the possible effects. Thus, the present study will contribute to the literature on digital

substance use prevention interventions among adolescents and young adults in several ways. The choice to study a digital intervention can from a public health perspective be regarded as positive. Digital brief interventions have several advantages over face-to-face approaches, for example, the reduction of stigma around help-seeking for substance use and easy dissemination to large groups of people, which makes them cost-effective. Moreover, digital interventions have previously proven effective in addressing alcohol and other substance use in the general population and some studies have shown promising results also among adolescents and young adults.[99 100] The current intervention is well grounded in theory and incorporates elements of MI and social influence theory that has been shown to be effective in reducing problematic substance use in prior research.[57] Importantly, there are also some limitations to the current study. One limitation concerns selection bias and thereby external validity, as recruitment requires either that individuals click on our advertisements at social media to be considered for inclusion, or that they attend upper secondary schools or have been in contact with existing clinics, aware of the opportunity to participate in the study. Thus, our participants may have certain personality traits, or are especially prone to help-seeking compared with a broader audience. The exclusion of people not understanding Swedish is also a limitation with regard to generalisability.

**Author affiliations**
[1]STAD (Stockholm Prevents Alcohol and Drug Problems), Centre for Psychiatry Research, Department of Clinical Neuroscience, Karolinska Institutet, & Stockholm Health Care Services, Region Stockholm, Stockholm, Sweden
[2]Centre for Psychiatry Research, Department of Clinical Neuroscience, Karolinska Institutet, & Stockholm Health Care Services, Region Stockholm, Stockholm, Sweden
[3]Department of Psychology, Uppsala University, Uppsala, Sweden
[4]Department of Pharmaceutical Biosciences, Uppsala University, Uppsala, Sweden

**Acknowledgements** The authors would like to thank Dr. Silke Diestelkamp and Dr. Christiane Baldus for invaluable help in the process of building the Swedish version of the WISEteens programme.

**Contributors** PK, AKS, JG and THE obtained funding for the study. PK, THE and CS further developed/modified the intervention. THE designed the study with contribution from PK, AKS, JG, AHB and CS. PK wrote this paper. All authors commented on successive manuscript drafts and approved the final version of the manuscript.

**Funding** The work was supported by the National Public Health Agency of Sweden (grant no. 02299-2016-6.2) and the Alcohol Research Council of the Swedish Alcohol Retailing Monopoly (grant no. 2018–0016). Author AHB was supported by the Swedish Research Council (grant no. K2012-61P-22131-01-6). The funding bodies had no role in study design, data collection, analysis, data interpretation or writing manuscripts.

**Competing interests** None declared.

**Patient and public involvement** Patients and/or the public were not involved in the design, or conduct, or reporting or dissemination plans of this research.

**Patient consent for publication** Not required.

**Provenance and peer review** Not commissioned; externally peer reviewed.

**ORCID iD**
Pia Kvillemo http://orcid.org/0000-0002-9706-4902

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
