## [Reviewer comments · BMJ Open]

ARTICLE DETAILS

TITLE (PROVISIONAL)	Effects of an automated digital brief prevention intervention targeting adolescents and young adults with risky alcohol and other substance use: Study protocol for a randomised controlled trial
AUTHORS	Kvillemo, Pia; Strandberg, Anna; Gripenberg, Johanna; Berman, Anne; Skoglund, Charlotte; Elgán, Tobias

VERSION 1 - REVIEW

REVIEWER	Adriana Sanudo UNIFESP-EPM
REVIEW RETURNED	30-Oct-2019

GENERAL COMMENTS	This is an original study protocol that plans a digital intervention among adolescents and young adults to prevent the use of alcohol and other substances. On my own review, I thought that this study protocol was well written, brief, and I also found very little to comment on regarding how to improve this work. Follow below my comments. 1. Why not considered binge drinking as another outcome? In the introduction the authors say about the importance of binge drinking ("some individuals drink more and binge drinking is still highly prevalent among young adults ...") so why not verify if the intervention reduce the binge drinking?2. The intervention, will be applied only at baseline? In the three and six months follow-up the participants will receive any feedback? Please explain.3. Instead of using na unrestricted randomisation I suggest using a stratified permuted block randomisation considering sex and age in the random allocation.4. Finally I want to know if the protocol paper is a planned study that will be started or an ongoing study, I didn't see in the manuscript the dates of the study. Overall, this is an interesting manuscript. I hope that responding to these inquiries will improve the manuscript further.
---

REVIEWER	Claire Garnett University College London, UK
REVIEW RETURNED	19-Nov-2019

GENERAL COMMENTS	This is an interesting paper reporting on a study protocol to assess the effectiveness of a brief digital intervention targeting prevention of
--

alcohol and other substance use among adolescents and young adults in Sweden. The proposed study is an important one though I believe the protocol would benefit from some clarifications and minor structural changes.

1. I think the introduction would be improved by having an additional final paragraph that sets up the study, why it's being done, why certain variables are being assessed as potential moderators, and the theoretical background to the intervention (currently in the methods and I think would be better placed in the introduction).

2. Please proof read the paper carefully as there are some spelling errors, missing punctuation (e.g. a missing bracket), and some sentences are too long (e.g. "Perfectionism will be measured by two subscales...") or have duplicate phrases in them.

3. Whilst this paper is a study protocol, it is not clear whether it is planned or currently ongoing. The dates of the study in terms of recruitment and data collection should be included in the protocol paper.

4. Consider using the PICOS format (participant-intervention-comparator-outcome-setting) for the title to clarify exactly what the intervention is being compared with and the outcome being measured.

Minor comments:

5. In the abstract, it would be helpful to include what tool will be used for screening risky substance use and what tool/measure is being used to assess the primary outcome measure.

6. In the abstract, I think the last sentence of the methods and analysis section 'The trial is expected...' would be better suited under ethics and dissemination.

7. In the article summary, perhaps include an example of 1 or 2 of the potential moderators being analysed.

8. In the introduction, the paper switches from talking about interventions via the internet to digital interventions, as there is a difference between these, perhaps either stick to one term or explain the difference between the two.

9. In the introduction, can you elaborate on what the "various results" relating to references 33, 38 and 39 are?

10. The 'Aim' section of the paper seemed too long to me, I think a lot of the information in there would be better suited to either the introduction or methods.

11. Under recruitment, please include more details in terms of the planned recruitment and data collection periods, what the contingency plans are if recruitment doesn't go to plan and mention the country/city/setting of where recruitment is occurring (i.e. the general population of...).

12. Is there any evidence that a movie ticket for each FU assessment enhances participation and FU rates? Perhaps worth looking at this paper: Watson et al. (2018):

<https://www.jmir.org/2018/8/e10351/>

13. The detailed rationale for why particular moderating variables are being measured would be better suited to the introduction.

14. Under measures, it would be best to select a single primary outcome (i.e. change in alcohol use at either one of the FU assessments, not both).

15. Please can you include the multiple options for all of the participant background data.

16. As mentioned in one of my major comments, I think the 'theoretical background of the intervention' would be better suited to the introduction so that the 'intervention' section of the methods just

	focuses on the description of the content. 17. It would also be helpful to have screenshots of the intervention, if possible. 18. Please include the rationale for why that particular control group was chosen. 19. Please specify which analysis (per protocol or ITT) will be the primary analysis and provide a rationale for that choice. 20. More details on the moderator analysis would be helpful, such as will there be post-hoc analyses stratified by any variable found to be moderating the effectiveness of the intervention? 21. Please include the reason for not including PPI in the design or planning of the study. Have you considered including them in the interpretation and/or dissemination of the results? 22. The details on the ethics could be briefer and perhaps link to PIS and/or consent form. 23. Do you have any plans on how to address the limitations identified or see if that is the case (e.g. participants prone to help-seeking compared with a broader audience)?
--	--

REVIEWER	Professor Adam R Winstock University College London and Global Drug Survey UK I have created a free digital app for alcohol use the drinks meter
REVIEW RETURNED	20-Nov-2019

GENERAL COMMENTS	This is an ambitious project which is well conceived. It is appropriately informed by past research and digital tools. The literature review and methodological grounding is sound. The plans and analysis and power calculation are also appropriate. First a few general questions. Do the researchers consider they will have the power to identify the most significant components of the feedback - ie which were considered most useful in shifting perceived norms or motivating a reduction in alcohol? Will there be any gender difference in feedback? if not why? I have 4 specific questions for the authors that pertain to 1) sampling 2) the possibility of unwanted consequences of the intervention (ie increased consumption) if the intervention was to be used widely within the normal population where there would be many people not at high risk; 3) the utility and calculation of blood alcohol levels and 4) to ask if they missed an opportunity to evaluate a short and long form of the intervention. 1 Sampling: It appears the recruitment process involves both accessing potential participants through existing services (high risk) and through social media. There is no mention whether the researchers will be able to identify the participants by recruitment strategy. I think this would be useful. For on line / social media engagement I would like to see some indication of the promotional strategy and whether an A/B design will be used to determine optimal approaches to engagement. 2 Possibility of negative effects. Although I understand the target group are higher risk consumers, if the trial were to prove positive I assume the intervention may be made available to a wider
---

	population including those not at high risk. The exclusion of lower risk individuals may mean that unforeseen negative consequences of the trial are not identified. for example, given that one of the interventions utilises normative feedback, it is possible that less frequent consumers might increase their use if they see their frequency of use is less than normal for example (depending on how feedback is provided). I wonder if the researchers considered this issue and whether excluding lower risk individuals from the trial in any way limits the utility of the study? 3 Predicted blood alcohol levels I would like some more information on the usefulness of feeding back on predicted blood alcohol levels to young people. I understand how the graphics would work, but it seems to me that how drunk people get subjectively would be a more meaningful measure than blood alcohol to young people. In addition heavier drinkers will be more tolerant to alcohol and might even see higher blood alcohol levels as positive (perverse 'badge of honour' if you will) . I would also ask for height and weight if you want to feedback on blood alcohol levels. I don't think one measure on their own works (I accept i might be wrong on this). 4 Long v shorter intervention analyses. At least some previous levels intervention trials (SIPS) have compared long and short forms of an intervention. although i understand additional recruitment is needed did the researchers considered have a brief version and long version?
--	---

VERSION 1 – AUTHOR RESPONSE

Reviewer(s)' Comments to Author:

Reviewer: 1

This is an original study protocol that plans a digital intervention among adolescents and young adults to prevent the use of alcohol and other substances. On my own review, I thought that this study protocol was well written, brief, and I also found very little to comment on regarding how to improve this work. Follow below my comments.

Thank you for the positive response.

1. Why not considered binge drinking as another outcome? In the introduction the authors say about the importance of binge drinking (“some individuals drink more and binge drinking is still highly prevalent among young adults ...”) so why not verify if the intervention reduce the binge drinking?

Thank you for the suggestion. We have now included binge drinking, as well as frequency of alcohol consumption and amount of alcohol consumed a typical day when alcohol is consumed, as secondary outcomes. See Abstract and page 7 under “Assessment”.

2. The intervention, will be applied only at baseline? In the three and six months follow-up the participants will receive any feedback? Please explain.

The participants will not receive any feedback after completing the program (directly after baseline), only instructions to fill out the two follow-up forms.

3. Instead of using an unrestricted randomisation I suggest using a stratified permuted block randomisation considering sex and age in the random allocation.

Thank you for an interesting suggestion. The randomization procedure is built into the program platform, making it difficult to stratify the allocation of participants. However, we think that the large number of participants will contribute to an evenly distribution between the study groups. We will also in statistical analyses control for sex/gender and age.

4. Finally I want to know if the protocol paper is a planned study that will be started or an ongoing study, I didn't see in the manuscript the dates of the study.

Thank you for the question. It is a planned study, the intervention has been built and will be pilot tested within the next couple of weeks. Recruitment will then start in early 2020, which is now described in the manuscript on page 6.

Overall, this is an interesting manuscript. I hope that responding to these inquiries will improve the manuscript further.

Thank you

Reviewer: 2

This is an interesting paper reporting on a study protocol to assess the effectiveness of a brief digital intervention targeting prevention of alcohol and other substance use among adolescents and young adults in Sweden. The proposed study is an important one though I believe the protocol would benefit from some clarifications and minor structural changes.

Thank you for the positive response.

1. I think the introduction would be improved by having an additional final paragraph that sets up the study, why it's being done, why certain variables are being assessed as potential moderators, and the theoretical background to the intervention (currently in the methods and I think would be better placed in the introduction).

Thank you for constructive advice, the manuscript has been revised accordingly with an extra paragraph after "Aim", see page 5:

Rationale and hypothesis

The current study will add to existing evidence regarding the effectiveness of digital interventions aimed at reducing alcohol and other substance use among young people. We hypothesize that participants in the intervention group will report reduced substance use (primarily alcohol consumption

at three-month follow-up), with a larger effect size compared to an active control group receiving health information about various substances. A number of potential moderators will be assessed to confirm or contradict previous research on what factors influences the development of substance use patterns over time and effects of prevention interventions in different target groups.

2. Please proof read the paper carefully as there are some spelling errors, missing punctuation (e.g. a missing bracket), and some sentences are too long (e.g. "Perfectionism will be measured by two subscales...") or have duplicate phrases in them.

Thank you for constructive advice, the manuscript has been proof read and revised accordingly.

3. Whilst this paper is a study protocol, it is not clear whether it is planned or currently ongoing. The dates of the study in terms of recruitment and data collection should be included in the protocol paper.

It is a planned study. The intervention has been built and will be pilot tested within the next couple of weeks. Recruitment will then start in early 2020 and is planned to be completed in October 2020. All results from follow-up measures will hopefully be completely received in May 2021. This process is now described in the manuscript on page 6.

4. Consider using the PICOS format (participant-intervention-comparator-outcome-setting) for the title to clarify exactly what the intervention is being compared with and the outcome being measured.

Thank you for this advice. We have now changed the title to:

"Effects of an automated digital brief prevention intervention targeting adolescents and young adults with risky substance use: Study protocol for a randomised controlled trial"

Minor comments:

5. In the abstract, it would be helpful to include what tool will be used for screening risky substance use and what tool/measure is being used to assess the primary outcome measure.

Thank you for the advice, the abstract is now revised accordingly.

6. In the abstract, I think the last sentence of the methods and analysis section 'The trial is expected...' would be better suited under ethics and dissemination.

Thank you for the advice, the abstract is now revised accordingly.

7. In the article summary, perhaps include an example of 1 or 2 of the potential moderators being analysed.

Thank you for the advice, we have now specified the moderators, see page 2.

8. In the introduction, the paper switches from talking about interventions via the internet to digital interventions, as there is a difference between these, perhaps either stick to one term or explain the difference between the two.

Thank you for this remark. In our view, delivery via the Internet (online) requires digital interventions but digital interventions can be delivered offline. We have tried to be more clear with the concepts, also changing “internet” to the word “online”, see page 3-5.

9. In the introduction, can you elaborate on what the “various results” relating to references 33, 38 and 39 are?

Thank you for the advice, some clarification has been made, see page 5:

“...with various results on e.g., substance use behaviours.”

10. The ‘Aim’ section of the paper seemed too long to me, I think a lot of the information in there would be better suited to either the introduction or methods.

Thank you for the advice, the manuscript is now revised accordingly, see page 2-4, 5-7

11. Under recruitment, please include more details in terms of the planned recruitment and data collection periods, what the contingency plans are if recruitment doesn’t go to plan and mention the country/city/setting of where recruitment is occurring (i.e. the general population of...).

Thank you for the advice, the recruitment-section is now revised accordingly, see page 6.

12. Is there any evidence that a movie ticket for each FU assessment enhances participation and FU rates? Perhaps worth looking at this paper: Watson et al. (2018):
<https://eur01.safelinks.protection.outlook.com/?url=https%3A%2F%2Fwww.jmir.org%2F2018%2F8%2Ffe10351%2F&data=02%7C01%7Cpia.kvillemo%40ki.se%7C2f32998680c9410998d908d774128f55%7Cbff7eef1cf4b4f32be3da1dda043c05d%7C0%7C0%7C637105495375862540&sdata=%2Fdps5%2BbTBsZyBPCBCuGPCViIL1D6ChC08ExEHL1WXDg%3D&reserved=0>

Thank you for this important question. Our experience from previous studies (unpublished) is that many participants see a value in a movie ticket. In the manuscript, we have now added that a cinema ticket corresponds to a value up to 15 Euro, see page 6.

13. The detailed rationale for why particular moderating variables are being measured would be better suited to the introduction.

Thank you for the advice, the manuscript is now revised accordingly, see page 2-3.

14. Under measures, it would be best to select a single primary outcome (i.e. change in alcohol use at either one of the FU assessments, not both).

Sorry for the confusion about this. The primary outcome is alcohol consumption at three-month follow-up, all others are secondary outcomes. This is now stated under "Assessment" on page 7:

Based on results from the previous European WISEteens study,³⁸ showing significant effects on alcohol consumption but not on other substance use, the primary outcome is alcohol consumption at three-month follow-up. Secondary outcomes include alcohol consumption at six-month follow-up, binge drinking, frequency of alcohol consumption, amount of alcohol consumed a typical day when alcohol is consumed at three and six months follow-up, other substance use, mental health, sexual risk behaviours, and perceived peer pressure at three and six-month follow-up.

15. Please can you include the multiple options for all of the participant background data.

Thank you for this remark, the multiple options for all background data are now included, see page 8:

Additionally, background data will be asked for using multiple choice questions: sex (man, women, other), age (15, 16...25), residence (name of municipality), occupation (secondary school, upper secondary school, university, not student, working, practicing, unemployed, other), school performance (merit value for students), and parents' education (secondary school, upper secondary school, university, don't know).

16. As mentioned in one of my major comments, I think the 'theoretical background of the intervention' would be better suited to the introduction so that the 'intervention' section of the methods just focuses on the description of the content.

Thank you for constructive advice, the manuscript has been revised accordingly, see page 3-5, 8-9.

17. It would also be helpful to have screenshots of the intervention, if possible.

Thank you for this suggestion. Screenshots are now provided as a supplementary file.

18. Please include the rationale for why that particular control group was chosen.

Thank you, we have now clarified this in one of the bullet points:

- An active control condition, blinded to the study condition, controls for expectation, detection and performance bias.

And also on page 12, under "Discussion of Strengths and limitations".

19. Please specify which analysis (per protocol or ITT) will be the primary analysis and provide a rationale for that choice.

Thank you for the question. ITT will be the primary analysis, which is now clarified on page 10, along with a rationale for the choice:

The reason for choosing ITT in the primary analysis is an ambition to keep the power of the study and also to avoid biased results due to selection of those completing all follow-ups, as they may have

special characteristics not representative for the whole study group, which may influence study results.

20. More details on the moderator analysis would be helpful, such as will there be post-hoc analyses stratified by any variable found to be moderating the effectiveness of the intervention?

Thank you for bringing this up. Yes, if we find moderator effects, we will carry out post-hoc analyses stratified by the detected variable. This is now clarified on page 10.

21. Please include the reason for not including PPI in the design or planning of the study. Have you considered including them in the interpretation and/or dissemination of the results?

PPI was not adopted in the design and planning of the current study, except for persons at the age of the target group reading manuscripts for the revised version of Wiseteens, and also participating in a pilot-test, which is now clarified on page 10 under "Patient and public involvement statement". The reason for not involving the target group was that a similar study of the program (Wiseteens) with a similar design (but with a wait-list control condition and slightly different moderators) has already been carried out and found to be feasible.

22. The details on the ethics could be briefer and perhaps link to PIS and/or consent form.

Thank you for constructive advice, the manuscript has been revised accordingly, see page 11.

23. Do you have any plans on how to address the limitations identified or see if that is the case (e.g. participants prone to help-seeking compared with a broader audience)?

Thank you for raising this issue, we think that a sufficient number of participants will be recruited from schools and social media, making the sample more comparable with the total population. Also, we control for help-seeking behaviour (moderator) in the analyses.

Reviewer: 3

This is an ambitious project which is well conceived. It is appropriately informed by past research and digital tools. The literature review and methodological grounding is sound. The plans and analysis and power calculation are also appropriate.

Thank you for the positive response.

First a few general questions.

Do the researchers consider they will have the power to identify the most significant components of the feedback - ie which were considered most useful in shifting perceived norms or motivating a reduction in alcohol?

Thank you for this question. As we are not collecting information on potentially mediating variables, we will not be able to identify significant components of feedback to shift perceived norms. Perhaps we will investigate this issue in a qualitative study later on.

Will there be any gender difference in feedback? if not why?

Yes, the response regarding BAC will be different depending on sex. The equation include a sex-specific factors for boys: .58 and for girls: .49. If participants report "other" as sex/gender the factor is .535. No other differences in feedback are provided with regard to sex.

I have 4 specific questions for the authors that pertain to 1) sampling 2) the possibility of unwanted consequences of the intervention (ie increased consumption) if the intervention was to be used widely within the normal population where there would be many people not at high risk; 3) the utility and calculation of blood alcohol levels and 4) to ask if they missed an opportunity to evaluate a short and long form of the intervention.

1 Sampling: It appears the recruitment process involves both accessing potential participants through existing services (high risk) and through social media. There is no mention whether the researchers will be able to identify the participants by recruitment strategy. I think this would be useful. For on line / social media engagement I would like to see some indication of the promotional strategy and whether an A/B design will be used to determine optimal approaches to engagement.

Thank you for raising this question. We will not recruit participants on alcohol/drug clinics specifically, so "high risk" participants will probably be evenly distributed across recruitment groups. All participants will be assessed/screened with regard to alcohol/drug consumption so that all included participants are (more or less) "risk drinkers" or drug users, see page 6

2 Possibility of negative effects. Although I understand the target group are higher risk consumers, if the trial were to prove positive I assume the intervention may be made available to a wider population including those not at high risk. The exclusion of lower risk individuals may mean that unforeseen negative consequences of the trial are not identified. for example, given that one of the interventions utilises normative feedback, it is possible that less frequent consumers might increase their use if they see their frequency of use is less than normal for example (depending on how feedback is provided). I wonder if the researchers considered this issue and whether excluding lower risk individuals from the trial in any way limits the utility of the study?

Thank you for the question, this is an important aspect. The intervention/program is not meant to be used by persons without risk behaviour related to alcohol or drugs, thereby lowering the risk of negative consequences. We agree that in case of abstainers or "low consumers" using the program, it might lead to increased consumption. However, this risk is difficult to totally avoid, but we think that the screening in the study context and also if/when the program will be provided for a larger audience, will minimize this risk.

3 Predicted blood alcohol levels I would like some more information on the usefulness of feeding back on predicted blood alcohol levels to young people. I understand how the graphics would work, but it seems to me that how drunk people get subjectively would be a more meaningful measure than blood

alcohol to young people. In addition heavier drinkers will be more tolerant to alcohol and might even see higher blood alcohol levels as positive (perverse 'badge of honour' if you will) . I would also ask for height and weight if you want to feedback on blood alcohol levels. I don't think one measure on their own works (I accept i might be wrong on this).

Length is not a factor in the equation, built into the current programme, so we do not ask for that.

We ask the person's weight to begin the interior of the Widmark equation (used to calculate eBAC), and the equation doesn't require information on length. There is of course a risk of a high eBAC being regarded as positive by some participants. However, in the feedback we handle this by reminding about the risks associated with a high eBAC. In a future study, it would be interesting to compare giving feedback based eBAC or based on number of glasses.

4 Long v shorter intervention analyses. At least some previous levels intervention trials (SIPS) have compared long and short forms of an intervention. although i understand additional recruitment is needed did the researchers considered have a brief version and long version?

Thank you for this interesting comment. That would be a very interesting future study, however, in the current study we were primarily interested in comparing the intervention group with an active control group, since the European study on Wiseteens, previously carried out, employed a design with a wait-list control group. Thereby we hope to expand the knowledge of programme effects.

VERSION 2 – REVIEW

REVIEWER	Adriana Sanudo UNIFESP-EPM
REVIEW RETURNED	06-Jan-2020

GENERAL COMMENTS	The authors answered the questions and I am pleased with the version presented. My suggestion is to accept for publication. Best regards!
---

REVIEWER	Claire Garnett University College London, UK
REVIEW RETURNED	10-Jan-2020

GENERAL COMMENTS	This is an interesting project and a well-written paper. I have some comments on how I believe it could be improved. My major comment relates to setting up the need for this RCT in the introduction. The authors mention the previous RCT of WISEteens and then mentions a modified version of this in the aims, but not why this current RCT is required, or how or why the intervention has been modified. The sample age group differs, though it's not clear if this is the only difference between interventions, and if so, why the original RCT would not generalise. Minor comments below: 1. In the abstract, please provide details on how the reduction in alcohol use (primary outcome) is being measured.
---

	2. The rationale and hypothesis refers to a “larger effect size” and then in recruitment, to “detect a small effect size” – please can you clarify this. 3. Can you include some brief details on how the advertising will be intensified if necessary. 4. What are the expected follow-up rates based on this type of follow-up strategy and incentive use. 5. The assessment section appears to be describing the outcome measures, I think a lot of this section would fit better under measures. 6. It is not clear how the primary outcome measure is being assessed – AUDIT-C or DDQ? 7. Appears to be a focus on alcohol in terms of primary outcome and the intervention (i.e. personalised feedback), perhaps it would be helpful if this was reflected in the title and abstract as “alcohol and other substance use”. 8. It would be great to see some screenshots of the intervention, or a link to where an interested reader could see them. Also, I would have found it helpful for the authors to have included a document detailing their responses to each of the original peer-reviewer comments.
--	--

REVIEWER	A Winstock University College London
REVIEW RETURNED	09-Jan-2020

GENERAL COMMENTS	I feel they have addressed my concerns in this revision
---

VERSION 2 – AUTHOR RESPONSE

Reviewer(s)' Comments to Author:

Reviewer: 1

Reviewer Name: Adriana Sanudo

Institution and Country: UNIFESP-EPM

Please state any competing interests or state 'None declared': None declared

Please leave your comments for the authors below The authors answered the questions and I am pleased with the version presented.

My suggestion is to accept for publication. Best regards!

Thank you

Reviewer: 3

Reviewer Name: A Winstock

Institution and Country: University College London Please state any competing interests or state 'None declared': None

Please leave your comments for the authors below

I feel they have addressed my concerns in this revision

Thank you

Reviewer: 2

Reviewer Name: Claire Garnett

Institution and Country: University College London, UK Please state any competing interests or state 'None declared': None declared.

Please leave your comments for the authors below This is an interesting project and a well-written paper.

I have some comments on how I believe it could be improved.

My major comment relates to setting up the need for this RCT in the introduction. The authors mention the previous RCT of WISEteens and then mentions a modified version of this in the aims, but not why this current RCT is required, or how or why the intervention has been modified. The sample age group differs, though it's not clear if this is the only difference between interventions, and if so, why the original RCT would not generalise.

Thank you for bringing our attention to this. The word "modification" involves new graphics and additional population based material regarding alcohol consumption, for the feedback given to participants in other age groups than the original. We have added some information on this on page 6:

"In this study, the effectiveness of a modified version of this intervention, with new graphics and population based feedback-generating data on alcohol consumption covering all included age groups, will be tested..."

In addition, although not connected to the original layout of the intervention, we wanted to test the intervention in a Swedish setting. The former RCT was carried out in four European countries in cooperation.

Minor comments below:

1. In the abstract, please provide details on how the reduction in alcohol use (primary outcome) is being measured.

Thank you for this remark. We have now added some information on this:

"The primary outcome is reduction in alcohol use, measured by AUDIT-C total score and average total drinks per week assessed by DDQ. "

2. The rationale and hypothesis refers to a "larger effect size" and then in recruitment, to "detect a small effect size" – please can you clarify this.

Thank you for this remark, we regret the unclear expression. What we intend is for the intervention group to report a significantly greater reduction in alcohol use than the control group, regardless of whether the difference is larger/greater or smaller. We have now reformulated this, please see p 6:

“We hypothesize that participants in the intervention group will report a significant reduction in substance use (primarily alcohol use), compared to an active control group receiving health information about various substances.”

and also p 11:

“To test the effectiveness of the intervention, we will assess whether participants in the intervention group to a greater extent report decreased substance use, sexual risk behaviours, perceived peer-pressure, and improved mental health after three and six months, respectively, compared to participants in the active control group”.

3. Can you include some brief details on how the advertising will be intensified if necessary.

Thank you for this suggestion. We have added some details on page 7 (underlined):

“If sufficient number of participants have not been recruited according to plan, we will extend the recruitment period and intensify the advertising by increasing visibility on social platforms.”

4. What are the expected follow-up rates based on this type of follow-up strategy and incentive use.

Thank you for this important question. In fully automated (or unsupported programs), where programs directed to people with depression have been studied with regard to dropout rates and reasons for this, rates around 75% have been found. We have added some information on this on page 11, based on an additional reference to a meta-analysis:

“In the current study, we assume a quite large dropout rate, based on previous research. For example, a meta-analysis comprising 17 studies of unsupported (fully automated) interventions for depression showed 74% dropout rate at post-treatment.”⁹⁷

5. The assessment section appears to be describing the outcome measures, I think a lot of this section would fit better under measures.

Thank you for this suggestion. We have now omitted the heading "Assessment", since there was very little text left under that heading after moving relevant text to the section "Measures" (p 7-8).

6. It is not clear how the primary outcome measure is being assessed – AUDIT-C or DDQ?

Thank you for bringing this up. Both instruments will be used to assess the primary outcome. Some clarification has not been made on page 8 (underlined):

"Alcohol use, including the primary outcome, will be measured using two instruments. One of them is the short version of the Alcohol Use Disorders Identification Test (AUDIT-C),⁷⁶ measuring frequency and amount of consumed alcohol and frequency of binge drinking, providing a widely used and valid index sum score for problematic alcohol use among adolescents.^{75 77} The other is the Daily Drinking Questionnaire (DDQ),⁷⁸ measuring a variety of parameters of alcohol use in a typical week. Previous research has demonstrated that the DDQ is highly correlated with other measures of self-reported alcohol consumption.⁷⁹ The primary outcome will be measured by AUDIT-C total score and average total drinks per week assessed by DDQ. The two primary outcome measures complement one another in that the AUDIT-C primarily offers an indication of hazardous or harmful use,⁸⁰ whereas the DDQ quantity measure will indicate the level of alcohol consumption in grams per week. Other substance use will be measured by a short version of the Drug Use Disorder Identification Test (DUDIT),⁸¹ including the first four items (DUDIT-C) which assess frequency of consumption..."

7. Appears to be a focus on alcohol in terms of primary outcome and the intervention (i.e. personalised feedback), perhaps it would be helpful if this was reflected in the title and abstract as "alcohol and other substance use".

Thank you for this remark. We have now changed the title to:

"Effects of an automated digital brief prevention intervention targeting adolescents and young adults with risky alcohol and other substance use: Study protocol for a randomised controlled trial".

We have also changed the wordings in the abstract:

"The current study will test the effectiveness of a fully automated digital brief intervention aimed at reducing alcohol and other substance use in adolescents and young adults aged 15-25 years."

8. It would be great to see some screenshots of the intervention, or a link to where an interested reader could see them.

Thank you for this suggestion. In our previous revision we added two screenshots from the program (uploaded), but it seems like they couldn't be seen by reviewers.

Also, I would have found it helpful for the authors to have included a document detailing their responses to each of the original peer-reviewer comments.

Thank you for this suggestion. We have now uploaded both the current response and the previous response in word files.

VERSION 3 - REVIEW

REVIEWER	Claire Garnett University College London, UK
REVIEW RETURNED	20-Feb-2020

GENERAL COMMENTS	Thank you for the detailed responses to my previous comments. I have one outstanding point related to the primary outcome and the two measures. The authors have now clarified that the primary outcome measure is being assessed by both AUDIT-C and DDQ. It is recommended by CONSORT that only one primary outcome is used in a RCT due to “problems of interpretation associated with multiplicity of analyses” (http://www.consort-statement.org/checklists/view/32--consort-2010/80-outcomes). Furthermore, I would say that the AUDIT-C does focus on consumption (and that the full AUDIT indicates hazardous or harmful use). The 3 questions are on quantity and frequency of typical drinking, and heavy episodic drinking; so you can calculate an estimate grams of alcohol consumed per week using AUDIT-C. As such, perhaps AUDIT-C could be listed as the single primary outcome measure and the DDQ as a secondary outcome measure. Also, my apologies for failing to spot the authors’ responses to the original peer-reviewer comments!
---

VERSION 3 – AUTHOR RESPONSE

Response to reviewer

Reviewer: 2

I have one outstanding point related to the primary outcome and the two measures. The authors have now clarified that the primary outcome measure is being assessed by both AUDIT-C and DDQ. It is recommended by CONSORT that only one primary outcome is used in a RCT due to “problems of interpretation associated with multiplicity of analyses” (<https://eur01.safelinks.protection.outlook.com/?url=http%3A%2F%2Fwww.consort-statement.org%2Fchecklists%2Fview%2F32--consort-2010%2F80-outcomes&data=02%7C01%7Cpia.kvillemo%40ki.se%7Ca5d23226373a428bbe5a08d7da400c1d%7Cbff7eef1cf4b4f32be3da1dda043c05d%7C0%7C0%7C637217840927274868&data=GtpjroiRDW2AptWj1G0UjsHjOuHes6NMIUJP1bkipQ%3D&reserved=0>). Furthermore, I would say that the AUDIT-C does focus on consumption (and that the full AUDIT indicates hazardous or harmful use). The 3 questions are on quantity and frequency of typical drinking, and heavy episodic drinking; so you can calculate an estimate grams of alcohol consumed per week using AUDIT-C. As such, perhaps AUDIT-C could be listed as the single primary outcome measure and the DDQ as a secondary outcome measure.

Thank you for this advice. We have revised the manuscript accordingly, please see page 1,7,8 and 11

Also, my apologies for failing to spot the authors’ responses to the original peer-reviewer comments!

No problem!